# Low replicability can support robust and efficient science

Stephan Lewandowsky [1,2]* & Klaus Oberauer [3]

There is a broad agreement that psychology is facing a replication crisis. Even some seemingly well-established findings have failed to replicate. Numerous causes of the crisis have been identified, such as underpowered studies, publication bias, imprecise theories, and inadequate statistical procedures. The replication crisis is real, but it is less clear how it should be resolved. Here we examine potential solutions by modeling a scientific community under various different replication regimes. In one regime, all findings are replicated before publication to guard against subsequent replication failures. In an alternative regime, individual studies are published and are replicated after publication, but only if they attract the community's interest. We find that the publication of potentially non-replicable studies minimizes cost and maximizes efficiency of knowledge gain for the scientific community under a variety of assumptions. Provided it is properly managed, our findings suggest that low replicability can support robust and efficient science.

[1] School of Psychological Science, University of Bristol, 12A, Priory Road, Bristol BS8 1TU, UK. [2] School of Psychological Science, University of Western Australia, Perth, WA, Australia. [3] Department of Psychology, University of Zurich, Zurich, Switzerland. *email: stephan.lewandowsky@bristol.ac.uk

Replicability is fundamental to science[1]. Any finding that cannot be replicated at best fails to contribute to knowledge and, at worst, wastes other researchers' time when they pursue a blind alley based on an unreliable result. The fact that the replicability of published findings is <30% in social psychology, hovers ~50% in cognitive psychology[2], and remains at ~67% even for studies published in *Nature* and *Science*[3], has therefore justifiably stimulated much concern and debate[4,5]. At one end of the spectrum, it has been suggested that failures to replicate are best considered as an interaction triggered by one or more (typically unknown) moderator variables that capture the idiosyncratic conditions prevailing during the original study, but that were absent during the replication attempt[6]. (An overview of this position can be found in ref. [1]). On this account, what matters is not whether an exact replication can reproduce the original effect but whether the underlying theory finds further support in conceptual replications (studies that use a variety of different manipulations or measures to operationalize the crucial theoretical variables)[6]. Contrary to this position, the success of independent replications is no greater for effects that were initially reported together with conceptual replications than effects that were reported in isolation[7].

At the other end of the spectrum is the view that low replicability arises for a number of reasons related to currently widespread—but suboptimal—research practices[8]. Several factors have been identified: (1) The use of small samples and the resultant low power of studies contributes to low replicability because the significant effect reported in an underpowered study is more likely to represent a type I error than the same effect obtained with a powerful study[9]. The harmful effects of low power can be amplified by questionable statistical practices, often referred to as *p*-hacking. (2) One form of *p*-hacking involves multiple sequential analyses that are used to inform further data collection. This process, known as the optional stopping rule[10], can lead to dramatic increases in type I error rates[11]. If applied repeatedly, testing of additional participants can guarantee a significant result under the null hypothesis if data collection continues until the desired *p* value is ultimately obtained. (3) Data are explored without differentiating between a priori hypotheses and post hoc reasoning. This is known as Hypothesizing After the Results are Known (HARKing) and, because the same data are used to identify a hypothesis as well as test it, HARKing renders the reported *p* values uninformative because they are known to be inflated[12]. (4) Publication bias in favor of significant results[13,14] amplifies the preceding three problems and additionally prevents the community from discovering when findings have failed to replicate.

Recommendations to avoid suboptimal research practices[15] and introduce transparency[5], such as through preregistration of method and analysis plan[16], more stringent significance levels[17,18], reliance on strong theories[19], or reporting all data irrespective of significance[20], therefore deserve support. Nonetheless, even flawless and transparent research may yield spurious results for the simple reason that all psychological measurements involve random variables and hence the possibility of type I errors. Spurious results can only be avoided if replications become a mainstream component of psychological research[1]. Highlighting the virtues of replications is, however, not particularly helpful without careful consideration of when, how, why, and by whom experiments should be replicated.

To examine those questions, we simulate an idealized and transparent scientific community that eschews *p*-hacking and other questionable research practices and conducts studies with adequate power (*P* = .8). We focus on an idealized community precisely because we wanted to examine the issues surrounding replication in the absence of contamination by questionable

research practices, although we also show later that our conclusions are robust to the injection of questionable practices and fraud. We measure the community's success (the number of correctly identified true phenomena that were of interest to the scientific community) and efficiency (the number of experiments conducted overall) under two different knowledge acquisition strategies and, orthogonally, two different replication regimes. The key attribute of our model is that not all findings are deemed to be equally interesting by the scientific community.

Knowledge acquisition is either discovery-oriented[21] or guided by theory (with the predictive merit of the theory being another design variable). Discovery-oriented research seeks to identify interesting findings by foraging across a wide landscape of possible variables and measures. On this approach, failure of any given experiment is uninformative because the underlying theory makes no exact predictions about specific phenomena, only that they should arise somewhere in the search space[19]. For example, researchers may look for instances in which people's representation of time, and its tacit link to future events and progress, can be primed by a bodily action. Exploration of various options may eventually reveal that turning a crank clockwise rather than anticlockwise primes people's openness to experience[22]. Discovery-oriented research is particularly vulnerable to producing nonreplicable results[23] because it relies on few constraints from theory or previous findings to select hypotheses for testing[24]. Moreover, because these hypotheses target eye-catching and counterintuitive findings that are a priori unexpected[24], the chance of testing a true hypothesis is low. Low prior probabilities of hypotheses, in turn, imply low replicability[21,24,25]. Theory-testing research, by contrast, focuses on a tightly constrained space of variables and measures for which the theory necessarily predicts an effect[19]. That is, predictions are tightly tethered to the theory, and falsification of a hypothesis provides more information than in discovery-oriented research. For example, if a theory of memory predicts that temporally isolated items should be recalled better than those that are temporally crowded, the fact that they are not (or only under some conditions) challenges the theory[26]. Conversely, if the theory has survived initial test, then it is unlikely to be completely off target. In consequence, the hypotheses that are chosen for further test have greater prior odds of being true, which in turn implies that positive findings are more likely to replicate[21].

In the simulation, the two knowledge acquisition regimes differ only in the manner in which true discoverable effects and the search for those effects are structured: for discovery-oriented research, both are random, whereas for theory-testing research, true effects are clustered together and the theory guides search in various degrees of proximity to the true cluster. For both regimes, 9% of all possible simulated experiments can discover a true effect ($P(H_1) = .09$). This value reflects estimates of the baserate of a psychological hypothesis being true[4].

Replication decisions are either private or public. Private replication decisions are modeled by investigators replicating any notable result and publishing only successfully replicated phenomena. Public replication decisions are modeled by investigators publishing any notable result, with the scientific community deciding which of those to replicate based on whether the results are deemed interesting. For discovery-oriented research, we consider only positive and significant results to be notable, because only the discovery of effects is informative[19]. By contrast, for theory-testing research, the discovery of reliable null effects is also informative—because they may falsify necessary predictions of a theory[19]—and in one of our simulations, we therefore also consider reliably established null effects to be notable candidates for replication.

Figure 1 contrasts the two replication regimes. In both regimes decision making is shared between individual investigators and the scientific community (represented by orange shading), and in both regimes the scientific community determines whether or not a finding is deemed interesting. The principal difference is whether the community selects interesting findings from among those known to have been replicated (private regime; Fig. 1a) or selects from a larger published set of studies of unknown r3eplicability and communally determines which of those studies deserve replication because of their potential interest (public regime; Fig. 1b).

Scientific interest is modeled on the basis of the observed citation patterns in psychology. Citations, by definition, are a good proxy for scientific interest, and expert judgment and analysis of replicability (Methods section) confirm that citations do not predict replicability. The actual distribution of citations is highly skewed, with nearly 40% of articles receiving five or fewer citations within 3 years of publication (Methods section) and only 1.3% of articles receiving >50 citations during that period. Very few articles receive high citations beyond common bibliometric time horizons[27], confirming that lack of citations indicates lasting lack of scientific interest. For the simulations, irrespective of replication regime, the probability of replication of a study increases with the number of citations. Specifically, we consider a finding to be interesting—and hence a candidate for replication—if its citation rate, obtained by random sample from the modeled distribution of citations, exceeds the 90th percentile of that distribution (Methods section). Varying degrees of sharpness of the decision were explored by varying the temperature parameter of a logistic decision function centered on the 90th percentile. Larger values of temperature imply a more graded threshold of scientific interest, rendering it more likely that articles with fewer citations are considered interesting. Figure 2 shows the distribution of citations in psychology together with the logistic threshold functions with the three values of temperature (1, 5, and 10) used in the simulations.

In summary, we model a scientific community under two different replication regimes. In one regime, all findings are replicated before publication to guard against subsequent replication failures. In the alternative regime, individual studies are published and are replicated after publication, but only if they attract the community's interest. The outcome measure of interest is the efficiency of knowledge generation; specifically, we consider the number of experiments conducted by the community overall that are required to discover a set number of true effects. To foreshadow, we find that the publication of potentially nonreplicable single studies minimizes cost and maximizes efficiency of knowledge gain for the scientific community.

## Results

**Discovery-oriented research**. Discovery-oriented research used a random selection of an independent and a dependent variable for each of the 100 studies simulated during the first round of experimentation. Because selection was random with replacement, multiple identical studies could be conducted and the same effect discovered more than once. This mirrors scientific practice

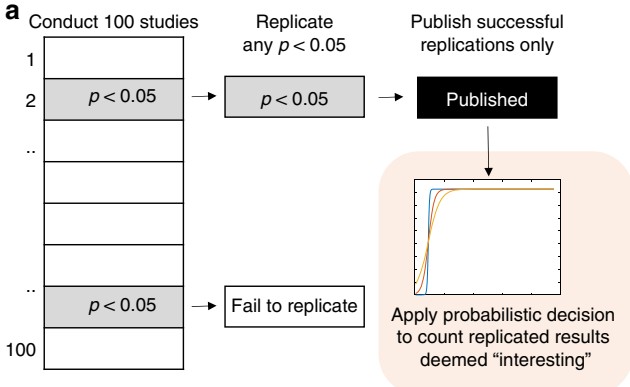

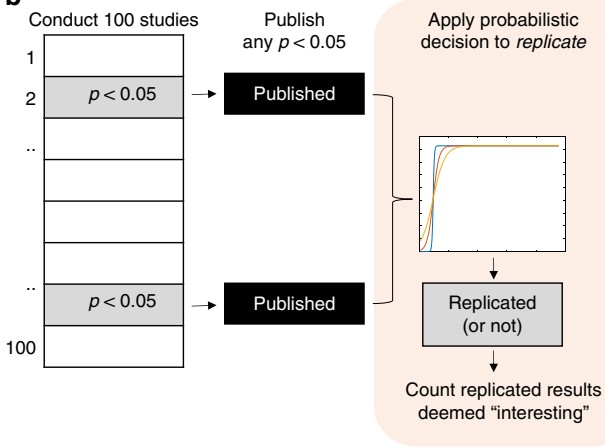

**Fig. 1 Comparison of two replication regimes. a** Private replication regime: investigators independently conduct 100 studies, and each investigator replicates any significant result. If replication is successful, both studies are published. The scientific community (represented by orange shading) then determines which of those replicated findings are deemed interesting based on a stochastic decision rule. **b** Public replication regime: investigators independently conduct 100 studies, and any significant result is published without replication. The scientific community (orange shading) then determines which of those findings of unknown replicability are deemed interesting, and hence worthy of replication, based on the same stochastic decision rule.

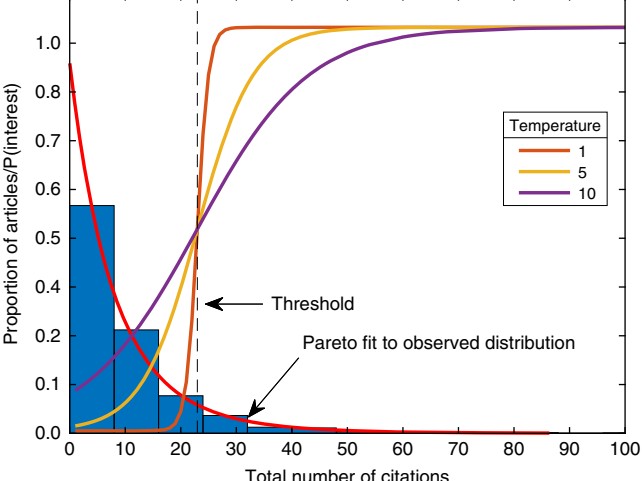

**Fig. 2 Distribution of citations for articles published in psychology.** The blue histogram shows observed distribution of citations for all articles published in 2014. The best-fitting Pareto distribution is represented by the solid red line. The threshold centered on the 90th percentile of the fitted distribution is indicated by the vertical dashed line. The three logistic functions are centered on the threshold but have different temperatures (see legend). They determine the probability of the scientific community finding a phenomenon to be of interest. In the simulation, each finding is associated with a random draw from the best-fitting Pareto distribution, which is then converted into a probability of interest using the appropriate logistic decision function.

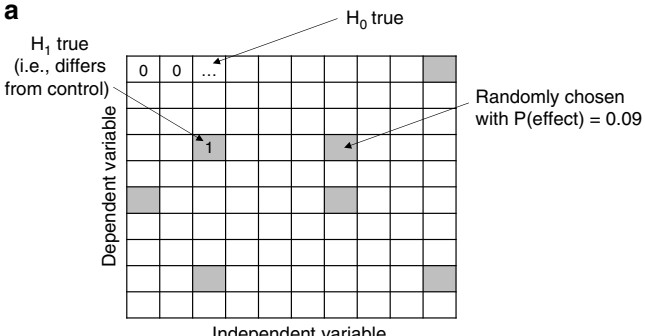

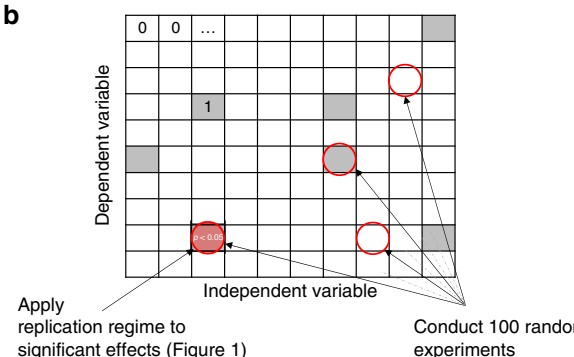

**Fig. 3 Illustration of the ground truth in discovery-oriented simulations.**
**a** Landscape of true effects for discovery-oriented research. Each cell is randomly initialized to 0 (i.e., $H_0$ is true) or 1 (i.e., $H_1$) with probability .09, reflecting the estimated baserate of true effects in psychology[4]. The landscape is initialized anew for each of the 1000 replications in each simulation experiment. **b** For each replication, 100 discovery-oriented experiments are conducted, each with a randomly chosen dependent and independent variable (sampling is with replacement). When an experiment yields a significant result ($p < .05$, two-tailed single-sample $t$-test or $BF_{10} > 3$, Bayesian single-sample $t$-test), the appropriate replication regime from Fig. 1 is applied.

that frequently gives rise to independent discoveries of the same phenomenon. Figure 3b provides an overview of the simulation procedure. Each study during the first round was classified as "significant" based either on its $p$ value ($p < .05$, two-tailed single-sample $t$-test) or its Bayes factor ($BF_{10} > 3$, Bayesian single-sample $t$-test with Jeffrey-Zellner-Siow prior, Cauchy distribution on effect size, see ref. [28]), irrespective of whether the null hypothesis was actually false. As is typical for discovery-oriented research, we were not concerned with detection of null effects. Some or all of the studies thus identified were then selected for replication according to the applicable regime (Fig. 1).

For frequentist analysis, we set statistical power either at 0.5 or 0.8. Figure 4 shows the results for the higher (Fig. 4a, b) and lower power (Fig. 4c, d). The figure reveals that regardless of statistical power, the replication regime did not affect the success of scientific discovery (Fig. 4b, d). Under both regimes, the number of true and interesting discovered effects increased with temperature, reflecting the fact that with a more diffuse threshold of scientific interest more studies were selected for replication in the public regime, or were deemed interesting after publication in the private regime. When power is low (Fig. 4d), fewer effects are discovered than when power is high (Fig. 4b). Note that nearly all replicated effects are also true: this is because the probability of two successive type I errors is small ($\alpha^2 = .0025$).

By contrast, the cost of generating knowledge differed strikingly between replication regimes (Fig. 4a, c), again irrespective of

statistical power. The private replication regime incurred an additional cost of around ten studies compared to public replications. This difference represents ~10% of the total effort the scientific community expended on data collection. Publication of single studies whose replicability is unknown thus boosts the scientific community's efficiency, whereas replicating studies before they are published carries a considerable opportunity cost. This cost is nearly unaffected by statistical power. Because variation in power has no impact on our principal conclusions, we keep it constant at 0.8 from here on. Moreover, as shown in Fig. 5, the opportunity cost arising from the private replication regime also persists when Bayesian statistics are used instead of conventional frequentist analyses.

The reasons for this result are not mysterious: Notwithstanding scientists' best intentions and fervent hopes, much of their work is of limited interest to the community. Any effort to replicate such uninteresting work is thus wasted. To maximize scientific productivity overall, that effort should be spent elsewhere, for example in theory development and test, or in replicating published results deemed interesting.

**Theory-testing research.** The basic premise of theory-testing research is that the search for effects is structured and guided by the theory. The quality or plausibility of a theory is reflected in how well the theory targets real effects to be tested. We instantiated those ideas by introducing structure into the landscape of true effects and into the experimental search (Methods section). Figure 6 illustrates the role of theory. Across panels, the correspondence between the location of true effects and the search space guided by the theory (parameter $\rho$) increases from 0.1 (poor theory) to 1 (perfect theory). A poor theory is targeting a part of the landscape that contains no real effects, whereas a highly plausible theory targets a segment that contains many real effects.

Not unexpectedly, the introduction of theory boosts performance considerably. Figure 7 shows results when all statistical tests focus on rejecting the null hypothesis, with power kept constant at 0.8. When experimentation is guided by a perfect theory ($\rho = 1$), the number of true phenomena being discovered under either replication regime with a diffuse decision threshold (high temperature) is approaching or exceeding the actual number of existing effects. (Because the same phenomenon can be discovered in multiple experiments, the discovery count can exceed the true number of phenomena.) The cost associated with those discoveries, however, again differs strikingly between replication regimes. In the extreme case, with the most powerful theory, the private replication regime required nearly 40% additional experimental effort compared to the public regime. The cost associated with private replications is thus even greater with theory-testing research than with discovery-oriented research. The greater penalty is an ironic consequence of the greater accuracy of theory-testing research, because the larger number of significant effects (many of them true) automatically entails a larger number of private replications and hence many additional experiments. As with discovery-oriented research, the cost advantage of the public regime persists irrespective of whether frequentist or Bayesian techniques are used to analyze the experiments.

There is nonetheless an important difference between the two classes of statistical techniques: Unlike frequentist statistics, Bayesian techniques permit rigorous tests of the absence of effects. This raises the issue of whether such statistically well-supported null results are of interest to the community, and if so, whether the interest follows the same distribution as for non-null results. In the context of discovery-oriented research, we assumed that null results are of little or no interest because failures to find

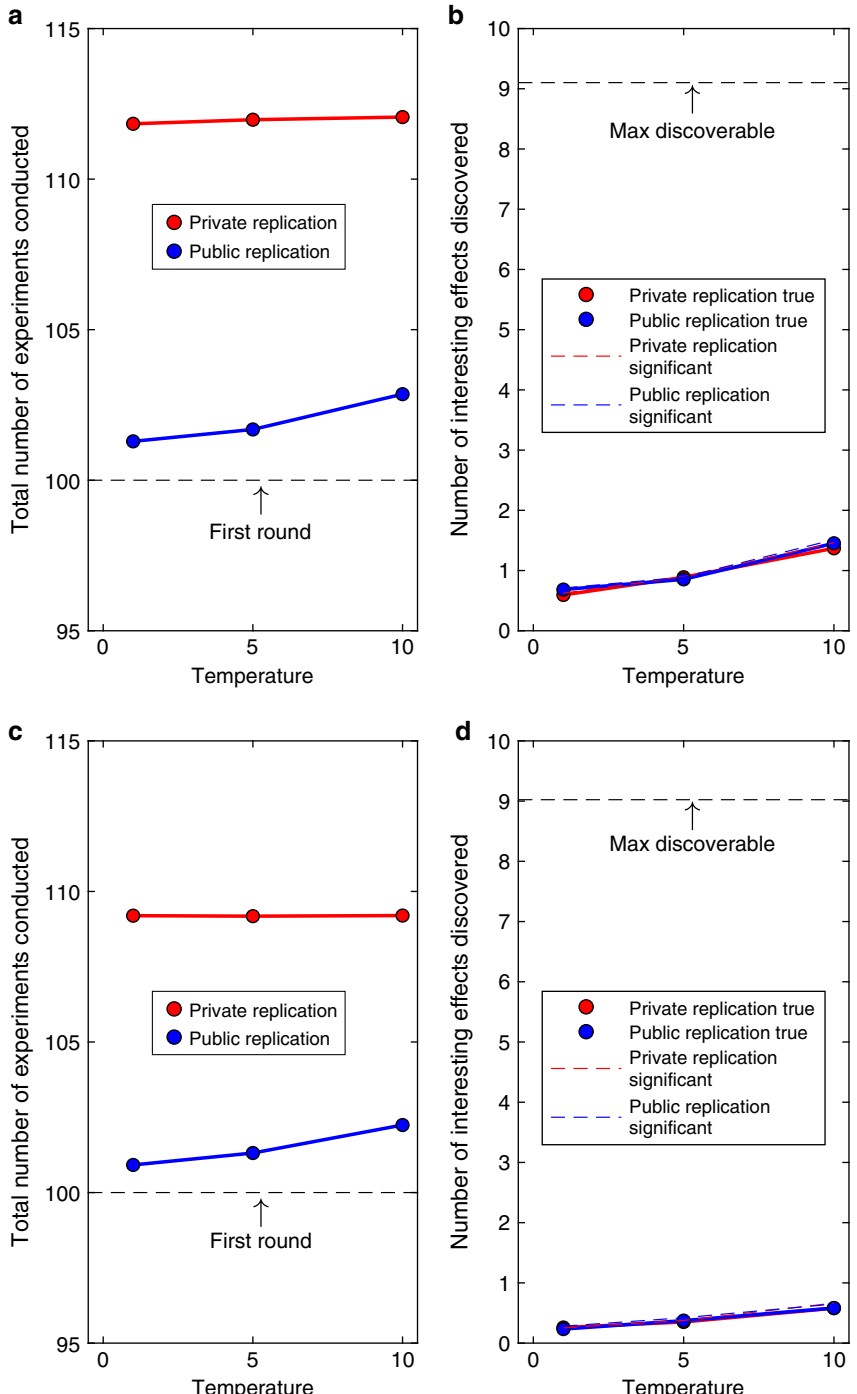

**Fig. 4 Scientific achievements and costs for discovery-oriented research, using frequentist analysis.** Power was 0.8 **a**, **b** or 0.5 **c**, **d**. **a** and **c** show the cost (total number of experiments conducted) to generate the knowledge (true effects discovered) shown in **b** and **d**. Temperature refers to the temperature of the logistic decision function (Methods section). Successful replications are identified by dashed lines in **b** and **d**. Successful replications that are also true (i.e., the null hypothesis was actually false) are identified by plotting symbols and solid lines.

an effect that is not a necessary prediction of any theory is of no theoretical or practical value[19]. The matter is very different with theory-testing research, where a convincing failure to find an effect counts against the theory that predicted it. We therefore performed a symmetrical Bayesian analysis for theory-testing research and assumed that the same process applied to determining interest in a null result as for non-null results. That is, whenever a Bayes Factor provided evidence for the presence of an effect $(BF_{10} > 3)$ or for its absence $(BF_{10} < 1/3 = BF_{01} > 3)$, we considered it a notable candidate for replication. Figure 8 shows

that when both presence and absence of effects are considered, the cost for the private replication regime is increased even further, to 50% or more. This is because there is now also evidence for null effects $(BF_{01} > 3)$ that require replication irrespective of whether they are deemed interesting by the community.

Another aspect of Fig. 8 is that the value of $\rho$ matters considerably less than when only non-null effects are considered. This is because a poor theory that is being consistently falsified (by the failure to find predicted effects) generates as many

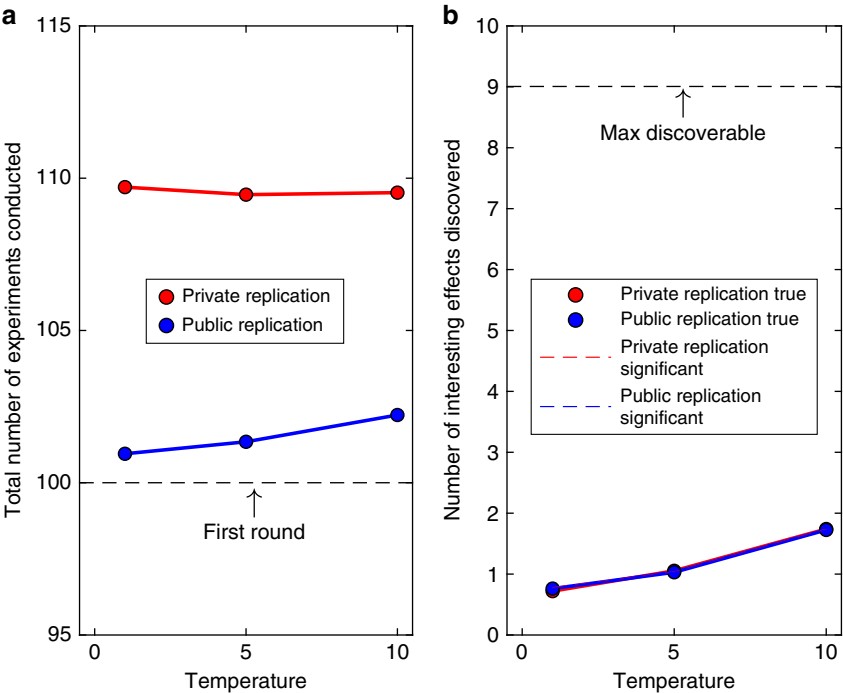

**Fig. 5 Scientific achievements and costs for discovery-oriented research, using Bayesian analysis. a** shows the cost (total number of experiments conducted) to generate the knowledge (true effects discovered) shown in **b**. Temperature refers to the temperature of the logistic decision function (Methods section). Successful replications are identified by dashed lines in **b**. Successful replications that are also true (i.e., the null hypothesis was actually false) are identified by plotting symbols and solid lines.

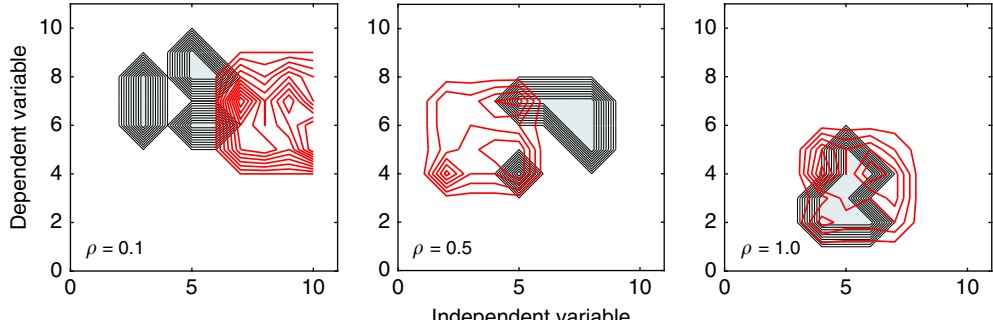

**Fig. 6 The role of theory in the model.** Each panel shows the same landscape of ground truth as in Fig. 3. In each panel, the gray contours illustrate the location of true effects (randomly chosen for each replication) and the red lines outline the space of experiments actually conducted as determined by the theory. Unlike in discovery-oriented research, true effects and experiments cluster together. The quality of the theory (determined by parameter $\rho$) is reflected in the overlap between the true state of the world (gray cluster) and the experiments (red cluster). The leftmost panel shows a poor theory ($\rho = 0.1$), the center panel a modestly powerful theory ($\rho = 0.5$), and the rightmost panel a perfect theory ($\rho = 1.0$). Each panel represents a single arbitrarily chosen replication.

interesting (null) results as a perfect theory that is consistently confirmed. Because our focus here is on empirical facts (i.e., effects and null-effects) rather than the welfare of particular theories, we are not concerned with the balance between confirmations and falsifications of a theory's predictions.

**Boundary conditions and limitations**. We consider several conceptual and methodological boundary conditions of our model. One objection to our analysis might invoke doubts about the validity of citations as an indicator of scientific quality. This objection would be based on a misunderstanding of our reliance on citation rates. The core of our model is the assumption that the scientific community shows an uneven distribution of interest in phenomena. Any differentiation between findings, no matter

how small, will render the public replication regime more efficient. It is only when there is complete uniformity and all effects are considered equally interesting, that the cost advantage of the public replication regime is eliminated (this result trivially follows from the fact that the public replication regime then no longer differs from the private regime). It follows that our analysis does not hinge on whether or not citation rates are a valid indicator of scientific quality or merit. Even if citations were an error-prone measure of scientific merit[29], they indubitably are an indicator of attention or interest. An article that has never been cited simply cannot be as interesting to the community as one that has been cited thousands of times, whatever one's personal judgment of its quality may be.

Another objection to our results might invoke the fact that we simulated an idealized scientific community that eschewed fraud

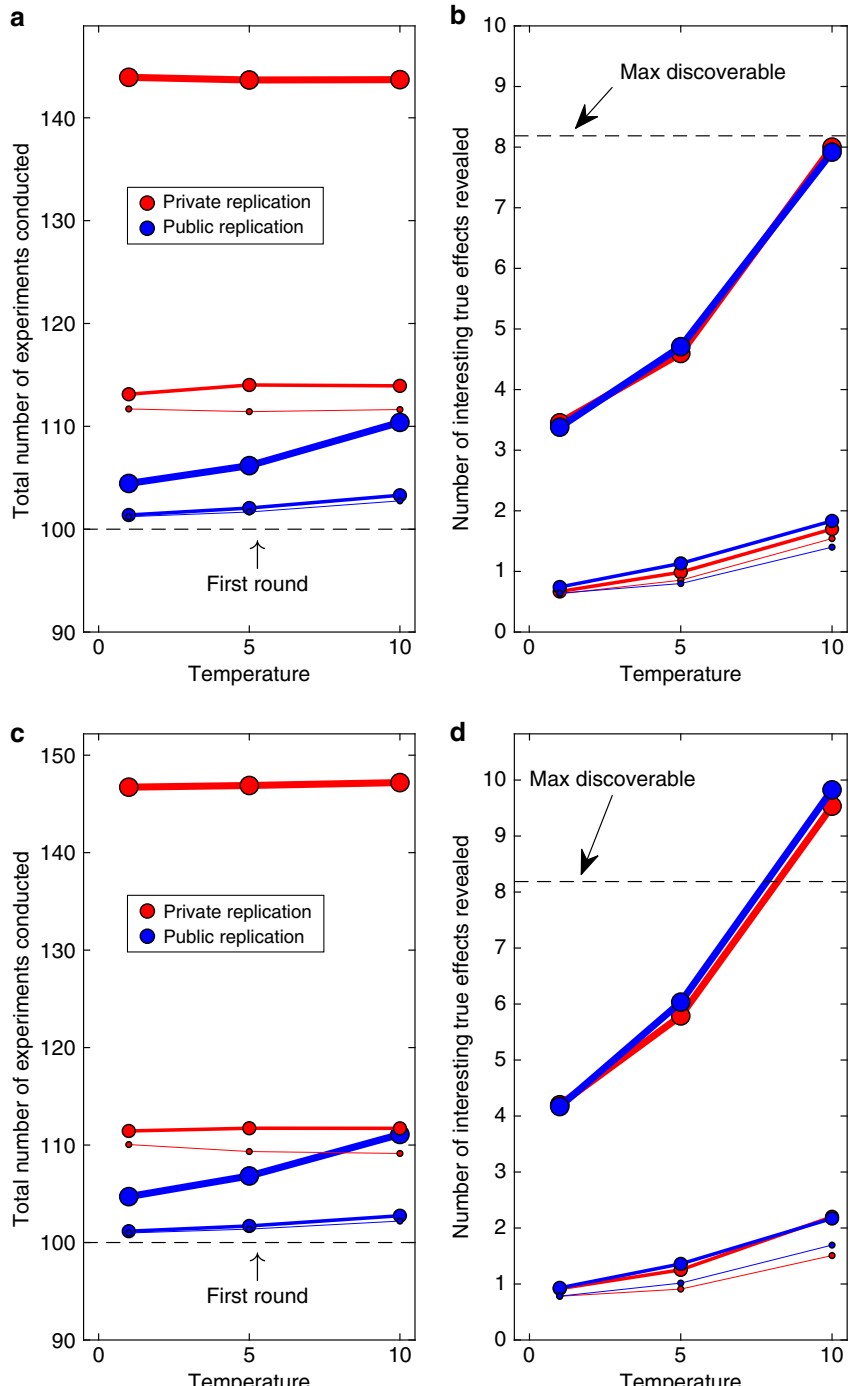

**Fig. 7 Scientific achievements and costs for theory-testing research.** The frequentist analysis is shown in **a** and **b**, and Bayesian tests for the presence of effects in **c** and **d**. **a** and **c** show the cost (total number of experiments conducted) to generate the knowledge (true effects discovered) shown in **b** and **d**. In all panels, the thickness of lines and size of plotting symbols indicates the value of $\rho$, which captures overlap between the theory and reality. In increasing order of thickness, $\rho$ was 0.1, 0.5, and 1.0. Temperature refers to the temperature of the logistic decision function (Methods section). All successful replications shown in **b** and **d** are true (i.e., the null hypothesis was actually false). Significant replications that did not capture true effects are omitted to avoid clutter.

or questionable research practices. We respond to this objection by showing that our model is robust to several perturbations of the idealized community. The first perturbation involves *p*-hacking. As noted at the outset, *p*-hacking may variously involve removal of outlying observations, switching of dependent measures, adding ad hoc covariates, such as participants' gender, and so on. A shared consequence of all those questionable research practices is an increased type I error rate: the actual $\alpha$

can be vastly greater than the value set by the experimenter (e.g., the conventional .05). Figure 9a, b shows the consequences of *p*-hacking with frequentist analysis, operationalized by setting $\alpha = 0.2$ in a simulation of discovery-oriented research. The most notable effect of *p*-hacking is that a greater number of interesting replicated effects are not true (difference between dashed and solid lines in Fig. 9b). The opportunity cost associated with private replications, however, is unaffected.

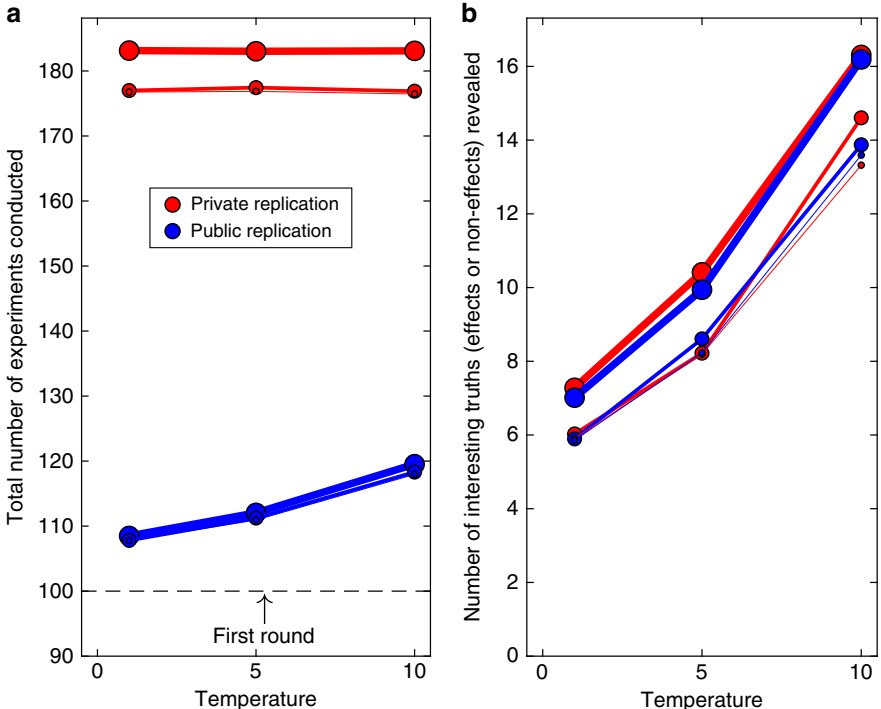

**Fig. 8 Scientific achievements and costs for theory-testing research, using a symmetrical Bayesian approach.** Tests are conducted for the presence ($BF_{10} > 3$) as well as absence ($BF_{01} > 3$) of effects. **a** shows the cost (total number of experiments conducted) to generate the knowledge (true effects and true absences of effects discovered) in **b**. Because both presence and absence of an effect are considered, the maximum discoverable number of true outcomes is 100. In both panels, the thickness of lines and size of plotting symbols indicates the value of $\rho$, which captures overlap between the theory and reality. In increasing order of thickness, $\rho$ was 0.1, 0.5, and 1.0. Temperature refers to the temperature of the logistic decision function (Methods section).

Figure 9c, d explores the consequences of an optional stopping rule, another common variant of *p*-hacking. This practice involves repeated testing of additional participants, if a desired effect has failed to reach significance with the initial sample. If this process is repeated sufficiently often, a significant outcome is guaranteed even if the null hypothesis is true[10]. We instantiated the optional stopping rule by adding $N_{ph} \in \{1, 5, 10\}$ additional participants, if an effect had not reached significance with the initial sample. This continued for a maximum of five additional batches or until significance had been reached. Optional stopping had little effect on the basic pattern of results, including the opportunity cost associated with the private replication regime, although persistent testing of additional participants, as expected, again increased the number of replicated results that did not represent true effects. Overall, Fig. 9 confirms that our principal conclusions hold even if the simulated scientific community engages in questionable research practices.

We examined two further and even more extreme cases (both simulations are reported in the online supplement). First, we considered the effects of extreme fraud, where all effects during the first round are arbitrarily declared significant irrespective of the actual outcome (Supplementary Fig. 3), and only subsequent public replications are honest (the private replication regime makes little sense when simulating fraud, as a fraudster would presumably just report a second faked significance level). Fraud was found to have two adverse consequences compared to truthful research: (a) it incurs a greater cost in terms of experiments conducted by other investigators (because if everything is declared significant at the first round, more effects will be of interest and hence require replication). (b) Fraud engenders a greater number of falsely identified interesting effects because all type I errors during the honest replications are assumed to represent successfully replicated findings. These results clarify

that our public replication regime is not comparable to a scenario in which completely fictitious results are offered to the community for potential replication—this scenario would merely mislead the community by generating numerous ostensibly replicated results that are actually type I errors.

Second, we considered the consequences of true effects being absent from the landscape of ground truths ($P(H_1) = 0$). This situation likely confronts research in parapsychology. In these circumstances, significant results from the first round can only reflect type I errors. In consequence, the overall cost of experimentation is lower than when true effects are present, but the cost advantage of the public regime persists (Supplementary Fig. 4).

## Discussion

Waste of resources has been identified as a major adverse consequence of the replication crisis[30]. We have shown that prepublication replications are wasteful. Perhaps ironically, waste is reduced by withholding replication until after publication. Regardless of whether research is discovery-oriented or theory-testing, and regardless of whether frequentist or Bayesian statistics are employed, the community benefits from publication of findings that are of unclear replicability. The cost advantage of the public replication regime was robust to various perturbations of the idealized community, such as *p*-hacking, fraud, and the pursuit of nonexistent effects.

Our model is consonant with other recent approaches that have placed the merit of research within a cost-benefit framework[18,31–33]. For example, Miller and Ulrich[33] examined the trade-off between false positives (type I errors) and false negatives (type II) under different payoff scenarios. Their model could determine the optimal sample size to maximize researchers' overall payoff, based on the recognition that although larger

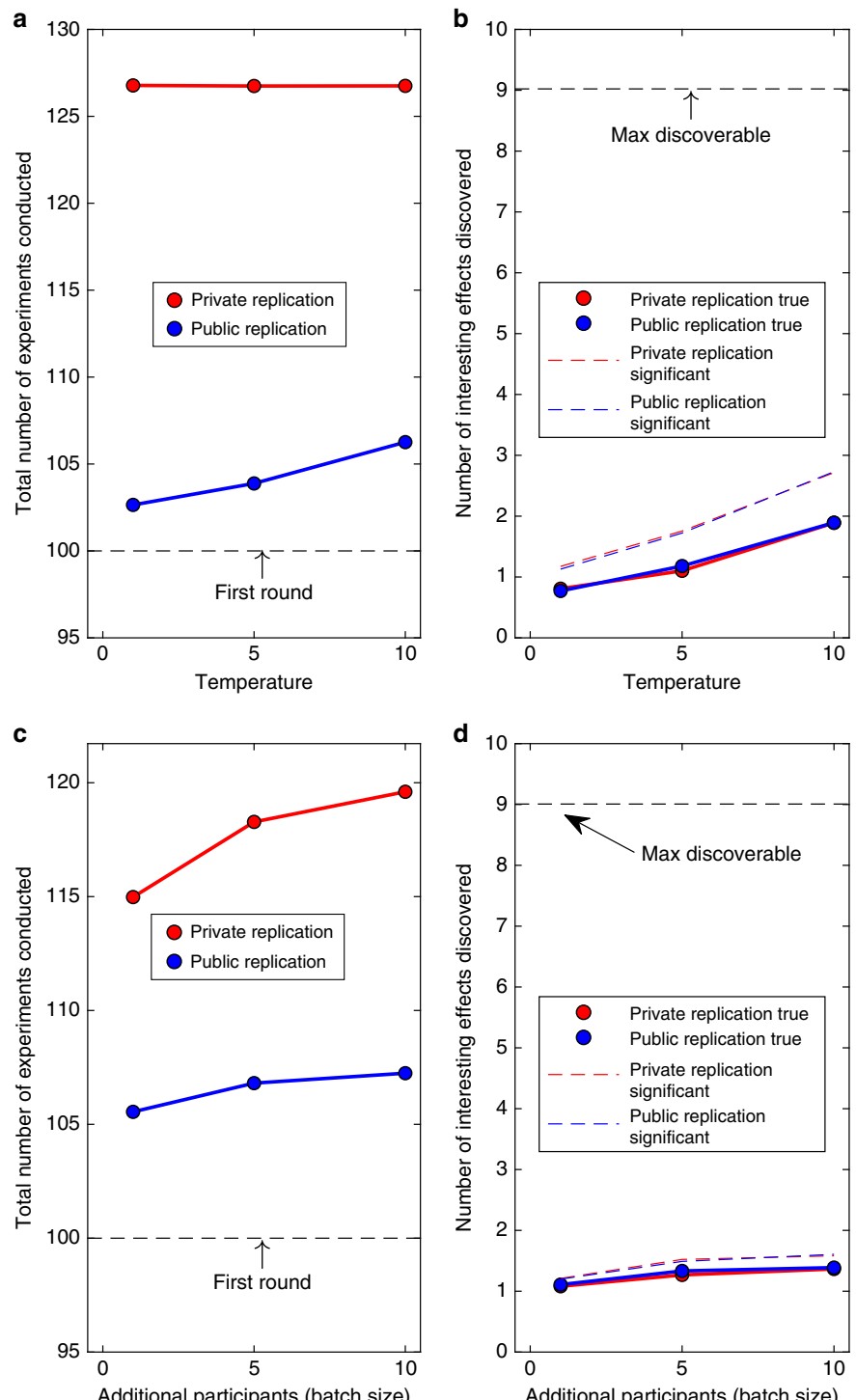

**Fig. 9 Effects of perturbations of the idealized scientific community. a** and **b** show the effects of *p*-hacking on the number of experiments conducted and the number of effects discovered. *p*-hacking is operationalized by raising the type I error rate, $\alpha = .2$. **c** and **d** show the effects of keeping $\alpha = .05$ but adding up to five batches of additional participants (each batch containing 1, 5, or 10 participants), if an effect failed to reach significance. Results are shown averaged across levels of temperature.

sample sizes increase power, they are also costly because they reduce the number of studies that can be carried out. In light of estimates that upward of 85% of research effort and resources are wasted because of correctable problems[34], any new practice that can free up resources for deployment elsewhere—e.g., to conduct advisable replications—should be given careful consideration.

Although we have shown private replications to be wasteful, adoption of our model would heed calls for a replication culture[35]

in several ways. Powerful and sophisticated replications require much investment[36], and by reducing the opportunity cost associated with unnecessary replications, our model frees up the resources necessary for powerful replications. Another favorable aspect of our model is that public replications are most likely conducted by laboratories other than the original investigator's. A recent expert survey (Methods section) revealed that 87% of experts considered a replication to be more informative if it is

conducted by a different lab, with the remainder (13%) considering replications by the same investigator to be equally informative. Out of >100 respondents, none thought that a replication by the original author was preferable to a replication by others. The overwhelming expert judgment is consonant with the result that replicability (by others) does not increase with the number of (conceptual) replications reported by the same author together with the original finding[7], and that too many low-powered replications in an article may reveal publication bias rather than indicate replicability[37,38]. In addition, our approach is entirely compatible with other solutions to the replication crisis, such as preregistration[16] or reliance on strong theory[19].

There are, however, some legitimate concerns that can be raised about a public replication regime. First, given that the regime is justified by greater efficiency of data collection, the increased burden on editors and reviewers that the regime implies through the increased number of publications is problematic. This added burden, however, is fairly modest by comparison to the gains in efficiency. To illustrate, for discovery-oriented research, the expected number of initially published findings under the public replication regime (with $P(H_1) = .09$, $\alpha = .05$, and power 0.8) is the expected number of significant results in the first round of experiments: $0.8 \times 9 + 0.05 \times 91 = 11.75$. This number is only slightly greater than the number of additional experiments carried out in the private replication regime (viz. the difference between private and public replication in Fig. 4a). Hence, factoring in the costs of editing and reviewing would render the public replication policy more costly only if the cost of reviewing and editing one publication is significantly larger than the cost of running one replication study. We maintain that this is rarely, if ever, the case: according to a recent detailed analysis of the global peer review system[39], each manuscript submission in psychology attracts 1.6 peer reviews on average, and the average duration to prepare a review is estimated at 5 h. It follows that total average reviewer workload for a manuscript is 8 h. Even if this estimate were doubled to accommodate the editor's time (for inviting reviewers, writing action letters and so on), the total additional editorial workload for the public replication regime would be $16 \times 11.75 = 188$ h. We consider it implausible to assume that this burden would exceed the time required to conduct the (roughly) ten additional replications required by the private regime. That said, careful consideration must be given to the distribution of workload: our analysis is limited to the aggregate level of the scientific community overall and does not consider potential inequalities across levels of seniority, gender, employment security, and so on. Our considerations here point to the broader need for a comprehensive cost-benefit analysis of all aspects of research, including replications under different regimes, that permits different payoffs to be applied to type I and type II errors[33]. However, this broader exploration goes beyond the scope of the current paper, in particular because the payoffs associated with statistical errors may vary with publication practice, as we discuss below.

A second concern arises from the perceived status of published nonreplicated results, which are an inevitable consequence of the public replication regime. It is likely that the media and the public would not understand the preliminary nature of such results, and even other researchers—especially when strapped for the resources required for replication—might be tempted to give undue credence to nonreplicated results. This is particularly serious for clinical trials, where a cautionary treatment of preliminary results is critical. Moreover, there is evidence that results, once published, are considered credible even if an article is retracted[40], and published replication failures seemingly do not diminish a finding's citation trajectory[41]. Hence, even if preliminary results are eventually subjected to public replication

attempts, a failure to replicate may not expunge the initial fluke from the community's knowledge base.

We take this concern seriously, but we believe that it calls for a reform of current publication practice rather than abandoning the benefits of the public replication regime. Adopting the public replication regime entails that published findings are routinely considered as preliminary, and gradually gain credibility through successful replication, or lose credibility when replications are unsuccessful. We suggest that the public replication regime can live up to its promise if (1) nonreplicated findings are published provisionally and with an embargo (e.g., 1 year) against media coverage or citation. (2) Provisional publications are accompanied by an invitation for replication by other researchers. (3) If the replication is successful, the replicators become coauthors, and an archival publication of record replaces the provisional version. (4) Replication failure leads to a public withdrawal of the provisional publication accompanied by a citable public acknowledgement of the replicators. This ensures that replication failures are known, thus eliminating publication bias. (5) If no one seeks to replicate a provisional finding, the original publication becomes archival after the embargo expires with a note that it did not attract interest in replication. This status can still change into (3) or (4) if a replication is undertaken later.

Although these cultural changes may appear substantial, in light of the replication crisis and wastefulness of current practice, cosmetic changes may be insufficient to move science forward. A recent initiative in Germany that provides free data collection for (preregistered) studies through a proposal submission process points in a promising direction (https://leibniz-psychology.org/en/services/data-collection/).

## Methods

**Simulation**. All simulations involved 1000 replications. The simulation comprised three main components.

The landscape of true effects was modelled by a $10 \times 10$ grid that represented the ground truth. For discovery-oriented research, the grid was randomly initialized for each replication to 0 ($H_0$) or 1 ($H_1$), with $P(H_1) = .09$ (Fig. 3a). The two dimensions of the grid are arbitrary but can be taken to represent potential independent and dependent variables, respectively. Each grid cell therefore involves a unique combination of an experimental intervention and an outcome measure, and the ground truth in that cell (1 or 0) can be understood as presence or absence, respectively, of a difference to a presumed control condition. For theory-testing research, the same landscape was used but all effects were randomly clustered within four rows and columns centered on a randomly chosen centroid (subject to the constraint that all effects fit within the $10 \times 10$ grid; Fig. 6).

The second component was a decision module to determine scientific interest. The distribution of citations for 1665 articles published in psychology in 2014 (downloaded from Scopus in April 2018) was fit by a generalized Pareto distribution (shape parameter, $k = 0.115$; scale parameter, $\sigma = 8.71$; and location parameter, $\theta = 0$; Fig. 2). For the simulations reported here, the 90th percentile of the fitted distribution ($q = 22.98$ citations) was used as threshold in a logistic transfer function:

$$P(I_k) = \frac{1}{1 + e^{-(n_k - q)/t}}, \tag{1}$$

where $P(I_k)$ is the probability that finding $k$ would be deemed interesting, $n_k$ represents the finding's citation count, and $t \in \{1, 5, 10\}$ the temperature of the logistic function. (The reciprocal of the temperature is known as the gain of the function.) Each $n_k$ represented a random sample from the best-fitting Pareto distribution. Other cutoff values of $q$ were explored, spanning the range from the 10th through the 90th percentile, which did not materially affect the results (Supplementary Figs. 1 and 2).

The final component was an experimental module to run and interpret experiments. Each simulation run (that is, each of 1000 replications) involved a first round of 100 experiments. Each experiment was simulated by sampling observations from a normal distribution with mean equal to the value of the targeted cell in the grid of ground truths (0 or 1) and standard deviation $\sigma$. The sample size was determined by G*Power[42] to achieve the desired statistical power. Power was either .5 or .8, mapping into sample sizes of $n = \{18, 34\}$. For frequentist analyses, $\sigma = 2.0$ and $\alpha = .05$ in all simulations. For Bayesian analyses, $n = 34$ and $\sigma = 1.5$ throughout, which achieved a "power" of ~0.8 with $BF_{10} = 3$. An experiment was declared "significant" if the single-sample $t$-statistic exceeded

**Table 1 Items, scale end points, and results summary of expert survey.**

| Item and (abridged) scale end points | Mean[a] | Prop = midpoint | Prop > midpoint |
|---|---|---|---|
| Which type of replication is most informative | | | |
| A direct (exact) replication—a conceptual replication | 6.46 | 0.39 | 0.28 |
| Replication by the same authors—replication in a different lab | 11.38* | 0.13 | 0.87 |
| All other variables being equal, which study is more likely to replicate? | | | |
| A rarely cited study—a widely cited study | 7.46 | 0.57 | 0.29 |
| A study that received no media attention—a study that received media attention | 6.0* | 0.46 | 0.12 |
| Result based on small sample ($N \simeq 20$)—result based on large sample ($N > 100$) | 10.11* | 0.19 | 0.72 |
| A study that was not pre-registered—a study with preregistered method | 10.13* | 0.23 | 0.70 |
| A study with preregistered method—a study with preregistered method and analysis plan | 10.95* | 0.15 | 0.81 |

[a] Asterisks indicate significant departure (maximum significant $p < .0007$, two-tailed single-sample $t$-test) from midpoint of scale (7)

the appropriate two-tailed critical value for $\alpha = .05$ or if $BF_{10} > 3$, Bayesian single-sample $t$-test as decribed in ref. [28].

For discovery-oriented research, the targeted cell in the landscape was chosen randomly (Fig. 3b). Theory-testing research also used a $10 \times 10$ grid to represent the gound truth, but all true effects (i.e., $H_1$) were constrained to fall within a $4 \times 4$ grid that straddled a randomly chosen centroid. For each simulated experiment, the targeted cell was chosen randomly from another $4 \times 4$ grid of predicted effects whose centroid was a prescribed distance from the centroid of true effects. The parameter $\rho$ determined the proximity between the centroid of true effects and the centroid of the predicted effects targeted by theory-testing research (Fig. 6). When $\rho = 1$, the centroids were identical, and for $\rho < 1$, the theory's centroid was moved $(1 − \rho) \times 9$ rows and columns away from the true centroid (subject to the constraint that all cells predicted by the theory had to fit within the $10 \times 10$ grid). A perfect theory ($\rho = 1$) thus predicted effects to be present in precisely the same area in which they actually occurred, whereas a poor theory ($\rho \simeq 0$) would search for effects in a place where none actually occurred.

The first round of 100 experiments was followed by replications as determined by the applicable regime (Fig. 1). Thus, under the private regime, any significant result from the first round was replicated, whereas under the public regime, significant results were replicated with a probability proportional to their scientific interest as determined by Eq. (1). (In the simulation that also examined null effects, see Fig. 8, replication decisions were also based on Bayes Factors for the null hypothesis).

**Expert survey**. Attendees of a symposium on statistical and conceptual issues relating to replicability at the International Meeting of the Psychonomic Society in Amsterdam (May 2018) were given the opportunity to respond to a seven-item single-page survey that was distributed before the symposium started. Responses were collected after each talk until a final set of 102 responses was obtained.

Each item involved a quasi-continuous scale (14 cm horizontal line) with marked end points. Responses were indicated by placing a tick mark or cross along the scale. Responses were scored to a resolution of 0.5 cm (minimum 0, maximum 14, and midpoint 7). Items, scale end points, and summary of responses are shown in Table 1.

**Reporting summary**. Further information on research design is available in the Nature Research Reporting Summary linked to this article.

## Data availability

MATLAB code for the simulation and all results are available at https://git.io/fhHjg. A reporting summary for this Article is available as a Supplementary Information file.

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

## Acknowledgements

The authors do not have funding to acknowledge.

## Author contributions

S.L. wrote and conducted the simulations and wrote the first draft of the paper. S.L. and K.O. jointly developed and discussed the project throughout.

## Competing interests

The authors declare no competing interests.
