## [Peer Review File · Nature Communications]

Reviewers' Comments:

Reviewer #1:

Remarks to the Author:

Summary

The authors employ a number of Monte-Carlo simulations to analyze the efficiency of two research regimes on replicating significant results: (a) a private replication regime and (b) a public replication regime. According to the private regime, each researcher replicates his or her own study that yielded a statistically significant result. If the replication turns out to be significant again, the result is published. According to the public regime, a researcher publishes each study that produces a significant result without replicating it. For either regime, the research community replicates those published studies that are deemed interesting. The simulation results show that the public replications regime outperforms the private regime in maximizing knowledge while minimizing costs. The authors analyze these regimes under various assumptions (frequentist vs. Bayesian statistics, discovery-oriented vs. theory guided, p-hacking vs. no p-hacking, etc.).

Evaluation

This is a most valuable and novel contribution to the current debate on replication. Most studies about the "replication crisis" merely discuss statistical rules and usually make suggestions on how to increase the replication rate (e.g., to lower the alpha level). With a few exceptions, however, they rarely consider the cost-benefit tradeoff implied by these suggestions. The authors model the replication process and thereby analyze this tradeoff for two plausible research regimes. In addition, their distinction between theory-oriented and discovery-oriented research is not only theoretically novel but also provides useful insights about these two approaches and their consequences with regard to successful replications.

To my mind, the assumptions of their simulation model are reasonable (of course, no model can ever be complete and yet still be useful; i.e. Bonini paradox). I think the authors achieved the right balance between model complexity and usefulness (yet, I believe that the complexity could be somewhat reduced in order to make model even more accessible to hasty readers, see Major Comment 2).

I strongly recommend that this paper is published. However, I also believe that it would benefit from some revisions. Specifically, the authors might consider my comments for improving the manuscript.

Major Comments

1. The paper might benefit from shortening the main text and relegating some points to the supplementary materials. For example, the text conceding the boundary conditions might be relegated to these materials.

2. I wonder if the assumptions and thus the simulations with regard to theory-oriented could be simplified. To my mind, $P(H1)$ is high when a theory is powerful. Therefore, why not run simulations without the 10×10 grid? I found the landscape notion -- that is associated with this grid -- difficult to comprehend. The authors might use $P(H1)=0.10$ for the discovery-oriented results, $P(H1) = 0.4$ for a modest powerful theory, and $P(H1)=0.7$ for a very powerful theory. In this case, no grid is required. See also Wilson and Wixted (2018, *Advances in Methods and Practices in Psychological Science*, 1, 186-197) on risky but potentially groundbreaking research.

3. The description from line 101 to 112 needs some improvement. It is too technical. One may just

say that the probability of replication of a study in the community will increase, the more often the study is cited. The somewhat long-winded justification of this approach seems not to be required.

Minor Comments

1. A tree or flow diagram of the simulation would be helpful.
2. The authors should mention in the introduction that a major factor for replicating a result is $P(H_1)$, that is, the base rate of true effects (for example, see Miller 2009, *Psychonomic Bulletin & Review*; Miller & Ulrich, 2019, *PlosOne*). I think that this issue must be emphasized because the base rate of true effects is the most important factor of replication probability. This would also help to better understand why the theory-oriented approach outperforms the discovery-oriented approach.
3. Why $P(H_1) = 0.09$ and not simply 0.10 or 0.05?
4. Figure 3: Please use the same scaling for the Y-axis in Panels a and c.

Rolf Ulrich

Reviewer #2:

Remarks to the Author:

In this manuscript the authors discuss replication and publication bias, and present a simulation of two alternative (meta-scientific) scenarios bearing on the question of whether or not scientific fields should publish non-replicable findings. The topic is timely and important. However, the gist of the paper concerns the simulation and I am afraid that might be too limited to warrant publication at this point. The simulation misses some relevant issues (underpowered studies and p-hacking in particular) which limited my early enthusiasm of the paper.

Specific points follow.

Line 19-20. Replicability of often confused with reproducibility but the two terms are not synonymous. Please define and use the terms accurately.

Line 26-27. The Registered Replication Report studies and the Many Lab studies in psychology offer very little evidence in favour of "hidden moderator" effects, pointing more strongly at the likelihood that biases are pervasive in many scientific literatures.

Lines 38-40. Note that researchers could p-hack by choosing to run multiple small (and hence underpowered) studies (Bakker et al., 2012 *Perspectives*) which exacerbates the problems of p-hacking and publication bias. Furthermore, it is important to note that this not only raises the Type 1 error rate, but also inflates true effects (Ioannidis, 2008).

Line 48. It would be good to mention and discuss pre-registration here.

Line 52. A crucial point here is that we should also publish all relevant studies if one were to focus on replication (van Assen et al. 2014 *PLOS ONE*), and that replication can actually make matters worse if we continue to put "failed" studies in the file drawer (Nuijten et al., 2015 *rev gen psych*).

Line 55. There is quite strong evidence to suggest that these assumptions do not hold in psychology.

Median power has long been known to be lower than .50 as suggested by several large scale meta-meta-analyses (Button et al., 201 Szucs & Ioannidis, 2017 PLOS BIO, Stanley et al., 2018 Psy Bull). And there is both direct and indirect evidence showing that p-hacking and publication bias are very common (from QRP surveys, misreported results, outcome switching, selective reporting, publication bias estimates from registers etc. etc.).

Line 84. There are many clear disadvantages to selective publication based on significance that should be discussed here (see van Assen et al. 2014 and de Winter & Happee 2013 PLOS ONE for a discussion on this). One major one is efficiency. Any loss of information might lead to the misdirected investment into false or otherwise inflated effects, or to research waste (Ioannidis et al, 2014 Lancet).

Lines 97-102. It is very important to address citation bias here: positive findings appear to be cited more than other -equally relevant- findings. Also the selection mechanisms of high impact factor journals -sometimes openly and deliberately focused on novelty and the risks for biases that go with it- should not be ignored in the discussion or the simulation.

Lines 135-139. The set-up of the simulation fails to address many potentially relevant characteristics of contemporary incentive structures (e.g., novel findings yielding more funding), research practices (e.g., the focus on "expanding" earlier results by conceptual replication rather than direct replication), research practices (p-hacking), and publication practices (failure to either submit or publish negative results).

Lines 140-143. I do side with the authors that theory-guided research lowers problems in scientific fields. They should perhaps relate this to Ioannidis' (2005) seminal work.

Lines 156-168. It is certainly valuable to focus on Bayesian techniques, but it is also crucial (from any perspective) to address the issue of power/sample size. The relation between N and the current results is not clear, while N per study is a very relevant part of the scientific system with a host of complexities. To wit, if earlier studies use small N approaches (which is very common and well documented) and entail false positives, later studies would require the use of appreciably larger N studies to "correct the record" (one could relate this to the statistical debate on the meaning of the reproducibility project psychology results). This should be a parameter in the simulation as well.

Lines 200-208. Extreme fraud (it is called scientific misconduct and should perhaps be specified as data fabrication in this context) is fairly uncommon (Fannelli, 2010 arrives at 2%) compared to the much more frequent use of QRPs in the analyses of data and reporting of results (probably > 50%). Hence, it would be more relevant to consider p-hacking/researcher degrees of freedom here.

Lines 214-223. It is crucial that any conclusion drawn is qualified by the (necessarily) limited set-up of the simulation. Hence, I would expect the authors to also discuss what they did not include in their study.

Reviewer #3:

Remarks to the Author:

I can be very brief. Like everyone I am quite interested in the general topic of this paper (how should we reform science?), but I just didn't see any strong and compelling results here that would merit publication in Nature Human Behavior. The basic problem is that in order to run these simulations

they have made a host of completely arbitrary assumptions. For every assumption they make, one can easily see a half dozen very different ways one might set it up. I also felt that the process that generated the figures showing up as the "cool" Figure 4 was exceptionally contrived, making the exercise pretty meaningless.

Response to Reviewers for NCOMMS-19-06872-T**Reviewer #1 (Remarks to the Author):**

Summary

The authors employ a number of Monte-Carlo simulations to analyze the efficiency of two research regimes on replicating significant results: (a) a private replication regime and (b) a public replication regime. According to the private regime, each researcher replicates his or her own study that yielded a statistically significant result. If the replication turns out to be significant again, the result is published. According to the public regime, a researcher publishes each study that produces a significant result without replicating it. For either regime, the research community replicates those published studies that are deemed interesting. The simulation results show that the public replications regime outperforms the private regime in maximizing knowledge while minimizing costs. The authors analyze these regimes under various assumptions (frequentist vs. Bayesian statistics, discovery-oriented vs. theory guided, p-hacking vs. no p-hacking, etc.).

Evaluation

This is a most valuable and novel contribution to the current debate on replication. Most studies about the "replication crisis" merely discuss statistical rules and usually make suggestions on how to increase the replication rate (e.g., to lower the alpha level). With a few exceptions, however, they rarely consider the cost-benefit tradeoff implied by these suggestions. The authors model the replication process and thereby analyze this tradeoff for two plausible research regimes. In addition, their distinction between theory-oriented and discovery-oriented research is not only theoretically novel but also provides useful insights about these two approaches and their consequences with regard to successful replications.

We are grateful for the reviewer's generally positive assessment.

To my mind, the assumptions of their simulation model are reasonable (of course, no model can ever be complete and yet still be useful; i.e. Bonini paradox). I think the authors achieved the right balance between model complexity and usefulness (yet, I believe that the complexity could be somewhat reduced in order to make model even more accessible to hasty readers, see Major Comment 2).

I strongly recommend that this paper is published. However, I also believe that it would benefit from some revisions. Specifically, the authors might consider my comments for improving the manuscript.

Major Comments

1. The paper might benefit from shortening the main text and relegating some points to the supplementary materials. For example, the text conceding the boundary conditions might be relegated to these materials.

On the basis of the Editor's action letter, which was based on additional communication with Reviewer 1, we have elected not to follow this recommendation.

2. I wonder if the assumptions and thus the simulations with regard to theory-oriented could be simplified. To my mind, $P(H1)$ is high when a theory is powerful. Therefore, why not run simulations without the 10 x 10 grid? I found the landscape notion -- that is associated with this grid -- difficult to comprehend. The authors might use $P(H1)=0.10$ for the discovery-oriented results, $P(H1) = 0.4$ for a modest powerful theory, and $P(H1)=0.7$ for a very powerful theory. In this case, no grid is required.

We disagree that a grid is not required if $P(H1)$ were manipulated directly. In our simulations, $H1$ is either true or false for a given simulated experiment: hence, $P(H1)$ can only be manipulated by varying the number of cells within the grid that are set to 1 (i.e., the number of experiments testing a true effect in the set of all possible experiments that could be run), and this of course requires the presence of a grid (or some other form of representing the set of possible experiments; nothing hinges on the 2-dimensional representation of that set). We hope that the new Figure 3, which summarizes our procedure and was requested by this reviewer, helps to illustrate the reasons for this more clearly.

See also Wilson and Wixted (2018, *Advances in Methods and Practices in Psychological Science*, 1, 186-197) on risky but potentially groundbreaking research.

We thank the reviewer for drawing our attention to this paper, which we now cite.

3. The description from line 101 to 112 needs some improvement. It is too technical. One may just say that the probability of replication of a study in the community will increase, the more often the study is cited. The somewhat long-winded justification of this approach seems not to be required.

We have augmented the description with a succinct sentence along the lines suggested by the reviewer. However, we had to retain the explanation of the role of temperature as it is a critical parameter in our simulations.

Minor Comments

1. A tree or flow diagram of the simulation would be helpful.

We have added a diagram (the new Figure 3), which together with Figures 1 and 2 visualizes the steps of the simulation. We hope that this new figure further clarifies why the grid is necessary to manipulate $P(H1)$.

2. The authors should mention in the introduction that a major factor for replicating a result is $P(H1)$, that is, the base rate of true effects (for example, see Miller 2009, *Psychonomic Bulletin & Review*; Miller & Ulrich, 2019, *PlosOne*). I think that this issue must be emphasized because the base rate of true effects is the most important factor of replication probability. This would also help to better understand why the theory-oriented approach outperforms the discovery-oriented approach.

We thank the reviewer for drawing our attention to those two articles, which we now cite. We agree that this is crucial and we have edited the introduction accordingly.

3. Why $P(H1) = 0.09$ and not simply 0.10 or 0.05?

We chose .09 on the basis of an empirical estimate (Dreber et al., 2015). While we appreciate the elegance of round figures, we felt that it was more important to mirror the best available estimate of the actual incidence of correct hypotheses.

4. Figure 3: Please use the same scaling for the Y-axis in Panels a and c.

Fixed. (Now Figure 4).

Reviewer #2 (Remarks to the Author):

In this manuscript the authors discuss replication and publication bias, and present a simulation of two alternative (meta-scientific) scenarios bearing on the question of whether or not scientific fields should publish non-replicable findings. The topic is timely and important. However, the gist of the paper concerns the simulation and I am afraid that might be too limited to warrant publication at this point. The simulation misses some relevant issues (underpowered studies and p-hacking in particular) which limited my early enthusiasm of the paper.

Specific points follow.

Line 19-20. Replicability of often confused with reproducibility but the two terms are not synonymous. Please define and use the terms accurately.

We have responded by using the word "replicability" (or derivatives) throughout.

Line 26-27. The Registered Replication Report studies and the Many Lab studies in psychology offer very little evidence in favour of "hidden moderator" effects, pointing more strongly at the likelihood that biases are pervasive in many scientific literatures.

We agree, and we take up that opposing position in the next paragraph. However, this paragraph is devoted to presenting the "hidden moderator" position without critique.

Lines 38-40. Note that researchers could p-hack by choosing to run multiple small (and hence underpowered) studies (Bakker et al., 2012 Perspectives) which exacerbates the problems of p-hacking and publication bias. Furthermore, it is important to note that this not only raises the Type 1 error rate, but also inflates true effects (Ioannidis, 2008).

We agree. We have added mention of the relationship between lack of power and p-hacking to this section.

Line 48. It would be good to mention and discuss pre-registration here.

Done. We now mention preregistration in several places where it is appropriate.

Line 52. A crucial point here is that we should also publish all relevant studies if one were to focus on replication (van Assen et al. 2014 PLOS ONE), and that replication can actually make matters worse if we continue to put "failed" studies in the file drawer (Nuijten et al., 2015 rev gen psyc).

We agree. We have added a discussion of those issues. In particular, our proposal for a new scheme for scholarly publication at the end of the paper has now been expanded to include a mechanism for failed replications to be put on record together with the original result.

Line 55. There is quite strong evidence to suggest that these assumptions do not hold in psychology. Median power has long been known to be lower than .50 as suggested by several large scale meta-meta-analyses (Button et al., 201 Szucs & Ioanidis, 2017 PLOS BIO, Stanley et al., 2018 Psy Bull). And there is both direct and indirect evidence showing that p-hacking and publication bias are very common (from QRP surveys, misreported results, outcome switching, selective reporting, publication bias estimates from registers etc. etc.).

This is certainly true, and we make no claim that our simulated scientific community is a realistic representation of current practice. We chose an idealized community for our simulation precisely because we want to flesh out the issues surrounding replication in the absence of "contamination" by questionable research practices. We now clarify our choice better, and we foreshadow here that we extend our simulations to cover some QRPs later in the paper.

Line 84. There are many clear disadvantages to selective publication based on significance that should be discussed here (see van Assen et al. 2014 and de Winter & Happee 2013 PLOS ONE for a discussion on this).

We agree, which is why one of our simulations addressed the issue of null effects and their replication (see Figure 8), although this consideration is restricted to theory-testing results. This restriction rests on the distinction between the first test of a new hypothesis and subsequent replication efforts. We agree – and say so in the revised Discussion – that replication efforts need to be published regardless of their outcome because otherwise publication bias would distort the state

*of published evidence on the hypothesis in question. However, if a first test of a hypothesis in the context of discovery-oriented research fails to find evidence for the hypothesized effect, that outcome is of virtually no scientific interest, because failures of hypotheses have no theoretical implications (we explore this in depth in a paper that is forthcoming in *Psychonomic Bulletin & Review*, and which we cite in the revision). Thus, withholding publication of such a null result does not contaminate the published record on this hypothesis. The worst possible outcome is that someone else repeats the futile effort, but that is unlikely because discovery-oriented research employs search in a wide, unconstrained space of possible hypotheses.*

One major one is efficiency. Any loss of information might lead to the misdirected investment into false or otherwise inflated effects, or to research waste (Ioannidis et al, 2014 *Lancet*).

We could not agree more—after all, maximizing efficiency by not wasting effort on replicating uninteresting results is at the heart of our model. However, this part of the manuscript is not the best place to discuss those issues; we take them up in the Discussion.

Lines 97-102. It is very important to address citation bias here: positive findings appear to be cited more than other -equally relevant- findings. Also the selection mechanisms of high impact factor journals -sometimes openly and deliberately focused on novelty and the risks for biases that go with it- should not be ignored in the discussion or the simulation.

Our simulation of discovery-oriented research captures the citation bias by focusing exclusively on positive findings.

Lines 135-139. The set-up of the simulation fails to address many potentially relevant characteristics of contemporary incentive structures (e.g., novel findings yielding more funding), research practices (e.g., the focus on “expanding” earlier results by conceptual replication rather than direct replication), research practices (p-hacking), and publication practices (failure to either submit or publish negative results).

We agree that our simulation does not address all of those issues—however, that was precisely the point of our work. As already noted, we intended to model an idealized scientific community to show that even under idealized conditions, pre-publication replications are not a panacea and, on the contrary, entail a loss of efficiency; see response to Line 55 above. Nonetheless, we have conducted some additional simulations; see response to Lines 200-208 below.

Lines 140-143. I do side with the authors that theory-guided research lowers problems in scientific fields. They should perhaps relate this to Ioannidis’ (2005) seminal work.

Done (albeit in the Introduction, where it seemed to fit better).

Lines 156-168. It is certainly valuable to focus on Bayesian techniques, but it is also crucial (from any perspective) to address the issue of power/sample size. The relation between N and the current results is not clear, while N per study is a very relevant

part of the scientific system with a host of complexities. To wit, if earlier studies use small N approaches (which is very common and well documented) and entail false positives, later studies would require the use of appreciably larger N studies to “correct the record” (one could relate this to the statistical debate on the meaning of the reproducibility project psychology results). This should be a parameter in the simulation as well.

We have complied with this suggestion and now report additional simulations in which this parameter is manipulated. Specifically, we vary power from .5 to .8 (in steps of .1) for discovery-oriented research. This turns out to have no effect on our principal conclusions.

Lines 200–208. Extreme fraud (it is called scientific misconduct and should perhaps be specified as data fabrication in this context) is fairly uncommon (Fannelli, 2010 arrives at 2%) compared to the much more frequent use of QRPs in the analyses of data and reporting of results (probably > 50%). Hence, it would be more relevant to consider p-hacking/researcher degrees of freedom here.

We note that we already reported results involving p-hacking in the immediately preceding section (old Figure 7a,b; Lines 186-192). The simulation reported the consequences of using an outcome-based stopping rule (i.e., test additional subjects until an effect is significant). Outcome-based stopping is a common instantiation of p-hacking.

In the revision, we conducted a further simulation in which the actual α was set to .20, but decisions about replications and conclusions continued to be drawn on the assumption that $\alpha=0.05$. This additional simulation serves to illustrate the distortions that are associated with QRPs generally: whatever the specifics of a QRP (e.g., switching outcome variables, excluding outliers), they all have in common the fact that the “garden of forking paths” during analysis is traversed in a manner that is dictated by a positive (i.e., significant) outcome. In consequence, QRPs generally increase α without the researcher’s awareness. This generic consequence of QRPs is captured in our new simulation.

Lines 214–223. It is crucial that any conclusion drawn is qualified by the (necessarily) limited set-up of the simulation. Hence, I would expect the authors to also discuss what they did not include in their study.

We agree and we have added a section to discuss those limitations.

Reviewer #3 (Remarks to the Author):

I can be very brief. Like everyone I am quite interested in the general topic of this paper (how should we reform science?), but I just didn't see any strong and compelling results here that would merit publication in Nature Human Behavior.

We note that the paper was actually submitted to Nature Communications.

The basic problem is that in order to run these simulations they have made a host of completely arbitrary assumptions. For every

assumption they make, one can easily see a half dozen very different ways one might set it up. I also felt that the process that generated the figures showing up as the "cool" Figure 4 was exceptionally contrived, making the exercise pretty meaningless.

We are unable to engage with this critique in depth as it provides too little detail. For example, we do not know what the "half dozen very different ways" are in which we could have set up our simulations.

Reviewers' Comments:

Reviewer #1:

Remarks to the Author:

The manuscript has clearly improved but I am still wondering about two points:

a) I still find the parameterization of the theory-driven research overcomplicated. A researcher relying on a plausible theory has simply a higher probability of testing for a real effect. I must admit that I do not understand the authors' response to my previous comment about this point.

b) I wonder how this analysis would relate to the payoff model by Miller & Ulrich (2016). To me, the authors seem to only consider the payoffs for true positives but ignore the ones for the other outcomes, especially with respect to the discovery-oriented research. A discussion of research payoffs is required.

Minor

Figure 4. There is a typo with respect to a,b,c, and d.

Reviewer #2:

Remarks to the Author:

The authors dealt well with my earlier concerns. This work is timely, important, and interesting. I have no further comments.

Response to Reviewers for NCOMMS-19-06872A

Reviewer #1 (Remarks to the Author):

The manuscript has clearly improved but I am still wondering about two points:

a) I still find the parameterization of the theory-driven research overcomplicated. A researcher relying on a plausible theory has simply a higher probability of testing for a real effect.

We agree entirely.

I must admit that I do not understand the authors' response to my previous comment about this point.

The idea that a plausible theory increases the probability of testing a real effect must be implemented in a simulation. We can think of no other way to achieve this than by creating a set of effects that can be tested, of which a subset is real. Once one accepts that premise (and the reviewer at least tacitly accepts this in the context of discovery-oriented research), then the plausibility of a theory must be expressed as the quality of the targeting of real effects to be tested. Hence the overlap between where the real effects are and where the theory is searching for them increases with the quality of the theory.

We chose to simulate this process using a set of 100 possible effects, which for illustrative purposes were arranged in a 10×10 grid because that makes it easy to visualize 100 possible effects. We could not find a way to simplify this simulation further. We have inserted a paragraph at the beginning of the theory-testing section that explains this in more detail.

b) I wonder how this analysis would relate to the payoff model by Miller & Ulrich (2016). To me, the authors seem to only consider the payoffs for true positives but ignore the ones for the other outcomes, especially with respect to the discovery-oriented research. A discussion of research payoffs is required.

Our analysis is on a less granular level than the payoff model of Miller and Ulrich: We simply assume a unit cost for carrying out an experiment, and benefits for the generation of knowledge (i.e., true, replicable effects, including true null effects), the size of which increases with the interest an effect attracts in the scientific community. We assign no value (no cost and no benefit) to false (i.e., non-replicable) findings. We acknowledge that a more sophisticated computational cost-benefit analysis could be set up if we assigned quantitative values to these costs and benefits. This would, however, make the analysis more complicated, and it would require fairly arbitrary assumptions about the quantitative benefits of generating knowledge, in comparison to the costs of running experiments. Moreover, it would need to make assumptions about how a revised publication culture, which we propose at the very end of the manuscript, would affect those costs and benefits. We believe that this would go much beyond the scope of the present manuscript. To

acknowledge the relation of our work to the payoff model of Miller & Ulrich, we have added a paragraph in the Discussion that discusses the Miller & Ulrich model in more detail.

Minor

Figure 4. There is a typo with respect to a,b,c, and d.

Oops. Fixed.

Reviewer #2:

The reviewer had no further comments.